# Social determinants of COVID-19 mortality at the county level

**Rebecca K. Fielding-Miller**[1]*, **Maria E. Sundaram**[2], **Kimberly Brouwer**[1]

**1** School of Medicine, University of California, San Diego, San Diego, California, United States of America,
**2** ICES Ontario, Toronto, Ottawa, Canada

* rfieldingmiller@health.ucsd.edu

**Data Availability Statement:** Analyses used publicly available data which are cited appropriately.

**Funding:** RFM was provided support by the National Institute of Mental Health, K01MH112436

## Abstract

As of August 2020, the United States is the global epicenter of the COVID-19 pandemic. Emerging data suggests that "essential" workers, who are disproportionately more likely to be racial/ethnic minorities and immigrants, bear a disproportionate degree of risk. We used publicly available data to build a series of spatial autoregressive models assessing county level associations between COVID-19 mortality and (1) percentage of individuals engaged in farm work, (2) percentage of households without a fluent, adult English-speaker, (3) percentage of uninsured individuals under the age of 65, and (4) percentage of individuals living at or below the federal poverty line. We further adjusted these models for total population, population density, and number of days since the first reported case in a given county. We found that across all counties that had reported a case of COVID-19 as of July 12, 2020 (n = 3024), a higher percentage of farmworkers, a higher percentage of residents living in poverty, higher density, higher population, and a higher percentage of residents over the age of 65 were all independently and significantly associated with a higher number of deaths in a county. In urban counties (n = 115), a higher percentage of farmworkers, higher density, and larger population were all associated with a higher number of deaths, while lower rates of insurance coverage in a county was independently associated with fewer deaths. In non-urban counties (n = 2909), these same patterns held true, with higher percentages of residents living in poverty and senior residents also significantly associated with more deaths. Taken together, our findings suggest that farm workers may face unique risks of contracting and dying from COVID-19, and that these risks are independent of poverty, insurance, or linguistic accessibility of COVID-19 health campaigns.

## Introduction

A novel coronavirus, SARS-CoV-2, is causing a global pandemic of COVID-19 respiratory disease. This pandemic has resulted in nearly 22 million cases and over 800,000 deaths since early January [1]. As of August 17 2020, the United States has more cases than any other nation in the world, with just over 5.4 million cases and 170,000 deaths [1]. Preliminary data indicates that existing health inequities in the United States are likely linked to COVID-19 morbidity and mortality [2].

and a National Institute of Minority Health and Health Disparities Loan Repayment Contract. The funders had no role in study design, data collection and analysis, decision to publish, or preparation of the manuscript.

**Competing interests:** The authors have declared that no competing interests exist.

Both infectious and non-communicable disease impact marginalized populations at disproportionate rates. While individual-level data is not currently available at the national level, data from county and state level entities suggest that COVID-19 may follow similar patterns. In the state of California, Latinos make up approximately 39% of the total population but represent just over 53% of total cases [3]. In New York City, Black/African American and Hispanic residents have significantly higher rates of COVID-19 illness and mortality than white residents, with a nearly doubled risk of mortality for Black/African American residents compared to white residents [4]. Journalistic reportings and early analyses suggest that unsafe working conditions among essential workers—who are more likely to be immigrants and/or racial/ethnic minorities [5]—and concerns about immigration status may be responsible for underlining existing health disparities among racial and ethnic immigrants across the United States, as well as potential language barriers, poverty, and a lack of insurance [6, 7].

We sought to assess the associations between COVID-19 mortality and farm worker, immigrant, and uninsured populations at the county level. We hypothesized that counties with a higher percentage of farm workers would report more deaths due to COVID-19, adjusting for poverty, insurance rates, population, age, and density.

## Methods

We built a series of spatial autoregressive models to assess county-level associations between the number of reported COVID-19 deaths in a county and the percentage of individuals engaged in hired farm work [8] in the county as of 2018. To account for potential confounders, we adjusted our analyses for the percentage of Non-English speaking households (defined as households in which no one 14 years or older reports speaking English at least "very well"), the percentage of uninsured individuals under the age of 65, percentage of individuals living at or below the poverty line, percentage of residents age 65 or older, county population, and county density, measured as the number of residents per square mile.

COVID-19 mortality data was sourced from county public health agencies, aggregated and made publicly available by the New York Times (NYT) [9]. For these analyses we used the NYT's "historical" dataset, with counts updated once per day with the final count of deaths as of that day [10]. Per the NYT:

> The data is the product of dozens of journalists working across several time zones to monitor news conferences, analyze data releases and seek clarification from public officials on how they categorize cases. . .At times, cases have disappeared from a local government database, or officials have moved a patient first identified in one state or county to another, often with no explanation. In those instances, which have become more common as the number of cases has grown, our team has made every effort to update the data to reflect the most current, accurate information while ensuring that every known case is counted.

Further details on the NYT's methodology as well as the full dataset is available at https://github.com/nytimes/covid-19-data. The proportion of households with limited English speaking ability was drawn from the American Community Survey's (ACS) 2014 5-year estimate [11]; percentages of individuals living below poverty and percentage of residents over the age of 65 were from 2017 ACS data [12, 13]. The percentage of farmworkers was taken from the US Bureau of Economic Analysis [14]. Percent uninsured was based on the US Census Small Area Health Insurance Estimates (SAHIE) program's 2018 estimates. Density was measured as the number of individuals per square mile, based on US census data. In addition to hypothesized predictors and potential confounders, we adjusted our models to account for the stage of

the local epidemic by including a variable for the number of days since the first case of COVID-19 was reported in a given county.

Counties with 1000 residents or more per square mile were coded as urban; counties with less than 1000 residents per square mile were coded as non-urban. While there are many ways to classify counties, we chose to use 1000 people per square mile for two reasons. First, the US census uses this threshold to designate census blocks as urban vs. non-urban [15]. Second, we felt that doing so allowed us to more clearly delineate major metropolitan areas and their associated resources and public health infrastructures from neighboring suburban or exurban counties.

All analyses were completed in Stata 16.0 (College Station, TX). We first built a series of simple linear regression models to assess the bivariate association between number of deaths within a county and our hypothesized predictors. We then constructed a spatial contiguity matrix and checked the assumption that residuals were distributed spatially using a Moran's I test.

We next built three separate spatial autoregressive models to assess the association between number of deaths and our hypothesized social determinants, adjusting for potential confounders, and fit the model with a spatial lag of the dependent variable based on our contiguity matrix. Our first model assessed relationships across all counties. We then stratified our analyses to measure the association between mortality and our hypothesized predictors in urban and non-urban counties. Our spatial autoregressive model used the generalized spatial two-stage least squares estimator, specifically Stata's spregress command with the "g2sl" option. G2sl is a generalized method of moment (GMM) modeling approach. The GMM makes no assumptions about variable distribution and is an appropriate statistical approach to address data that are highly skewed or have unknown distributions [16].

After assessing nation-wide trends, we conducted series of sub-analyses by US Census region within the contiguous 48 states, again stratifying by urban and non-urban counties. We first assessed the association between absolute number of deaths and the same predictors assessed in the nation-wide models. We then fit an additional series of models with the number of deaths per 100,000 residents as our primary outcome of interest, rather than the number of deaths in a county.

## Results

This analysis encompassed 3024 counties from all 50 states. As of July 12, 2020, the number of deaths reported in the New York Times' aggregated dataset ranged from 0 to 22,755 per county, with a median of 2 (IQR: 0–11) (Fig 1) Within this dataset, the 5 boroughs/counties of New York are treated as a single entity. We have done the same in these analyses, assigning all 5 counties the values associated with New York County. We classified 115 counties as urban and 2909 counties as non-urban. Deaths in urban counties ranged from 4–22,755, with a median of 272 and an IQR of 90–774. Deaths in non-urban counties ranged from 0–1,133 with a median of 2 and an IQR of 0–9 (Table 1). The number of deaths per 100,000 residents ranged from 0 to 350, with an overall median of 6 (IQR: 0–11) (Fig 2). The number of deaths per 100,000 was substantially higher in urban than non-urban counties. In urban counties the median was 39.1 per 100,000 (IQR: 15.0–107.3) while in non-urban counties it was 5.6 (IQR: 0–19.7). The geographic distributions of our primary predictors of interest are shown in Fig 3.

In our fully adjusted model with all 3024 US counties that had reported at least one case of COVID-19 as of July 12, 2020, the percentage of farm workers in a county, percentage of residents living at or below the federal poverty line, number of residents per square mile (ie, population density), and the percentage of residents over the age of 65 were all significantly

# COVID-19 deaths per county as of July 12, 2020

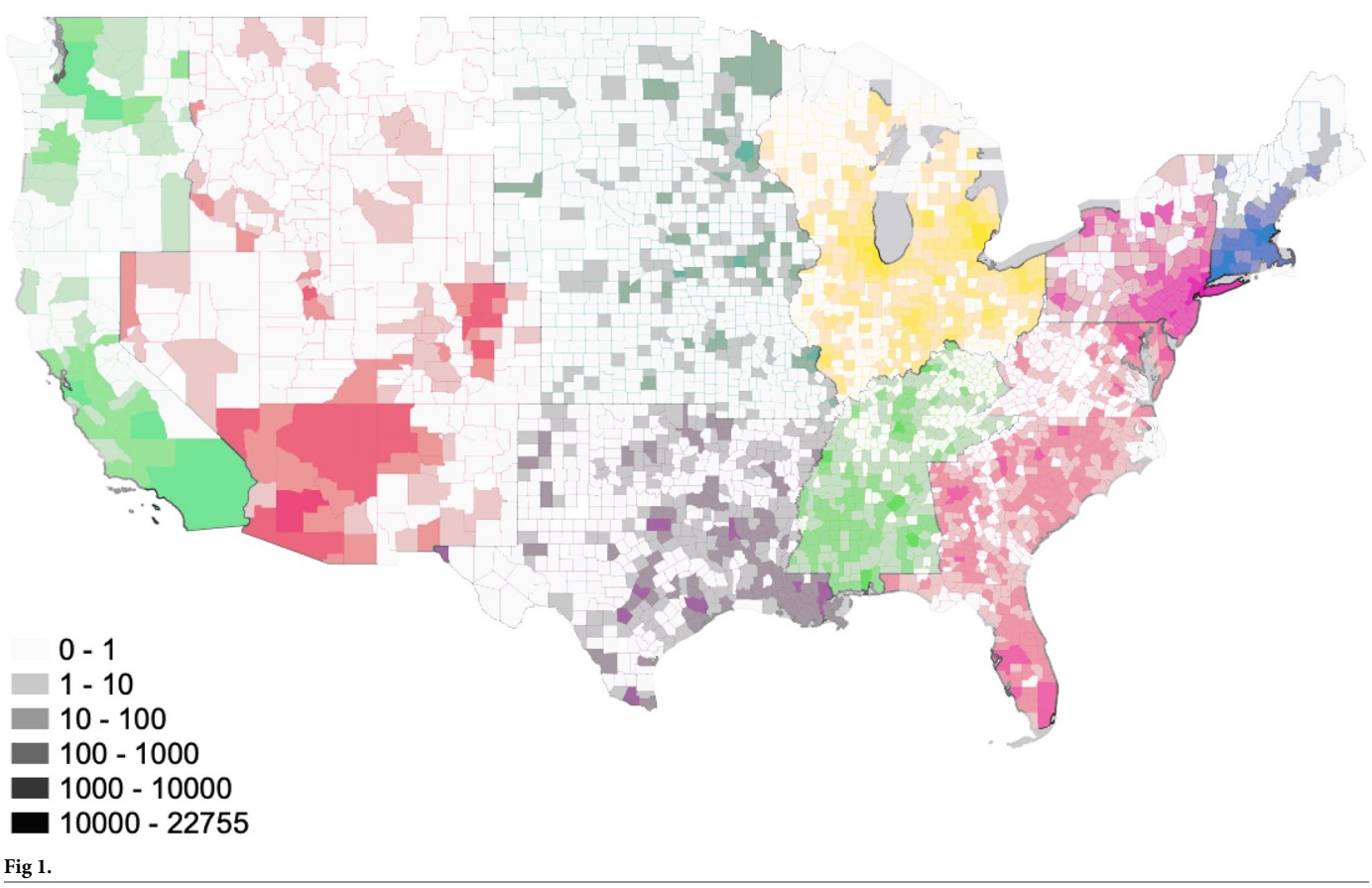

| | |
|---|---|
| ☐ | 0 - 1 |
| ☐ | 1 - 10 |
| ☐ | 10 - 100 |
| ☐ | 100 - 1000 |
| ☐ | 1000 - 10000 |
| ☐ | 10000 - 22755 |

**Fig 1.**

associated with a higher number of reported COVID-19 deaths (Table 2). Each additional percentage point of farmworkers in a county was associated with 5.79 more deaths (5.51 directly, 0.28 via indirect 'spillover' to the next county, p <0.001), while each additional percentage point of individuals living in poverty was associated with 4.41 additional deaths (4.20 directly,

**Table 1. Primary predictor and covariates of interest across all counties and stratified by urban and non-urban.**

| | All counties (n = 3024) | | Non-urban counties (n = 2909) | | Urban counties (n = 115) | |
|---|---|---|---|---|---|---|
| | *median* | *IQR* | *median* | *IQR* | *median* | *IQR* |
| *Number of Deaths* | 2 | 0–11 | 2 | 0–9 | 272 | 90.0–774.0 |
| *Deaths per 100,000* | 6 | 0–22 | 6 | 0–20 | 39 | 15–107 |
| *% Farm workers* | 2.3 | 0.9–4.9 | 2.4 | 1.1–5.0 | 0.1 | 0.0–0.1 |
| *% Non-English speakers* | 4.9 | 2.8.– 10.1 | 4.7 | 2.8–9.4 | 19.0 | 11.4–29.8 |
| *% Residents uninsured* | 10.6 | 7.4–14.6 | 10.7 | 7.5–14.6 | 8.3 | 5.8–12.6 |
| *% Residents in poverty* | 15.1 | 11.4–19.4 | 15.1 | 11.4–19.6 | 13.0 | 8.9–16.7 |
| *% Residents Over 65* | 16.4 | 13.9–19.0 | 16.6 | 14.2–19.1 | 12.5 | 11.0–14.5 |
| *Residents per square mile* | 45.5 | 17.9–111.9 | 42.7 | 17.0–98.6 | 1754.9 | 1313.4–2715.3 |
| *Population (thousands)* | 26.5 | 11.7–68.7 | 25.0 | 11.2–60.1 | 735.3 | 492.3–999.0 |

## COVID-19 deaths per 100,000 per county as of July 12, 2020

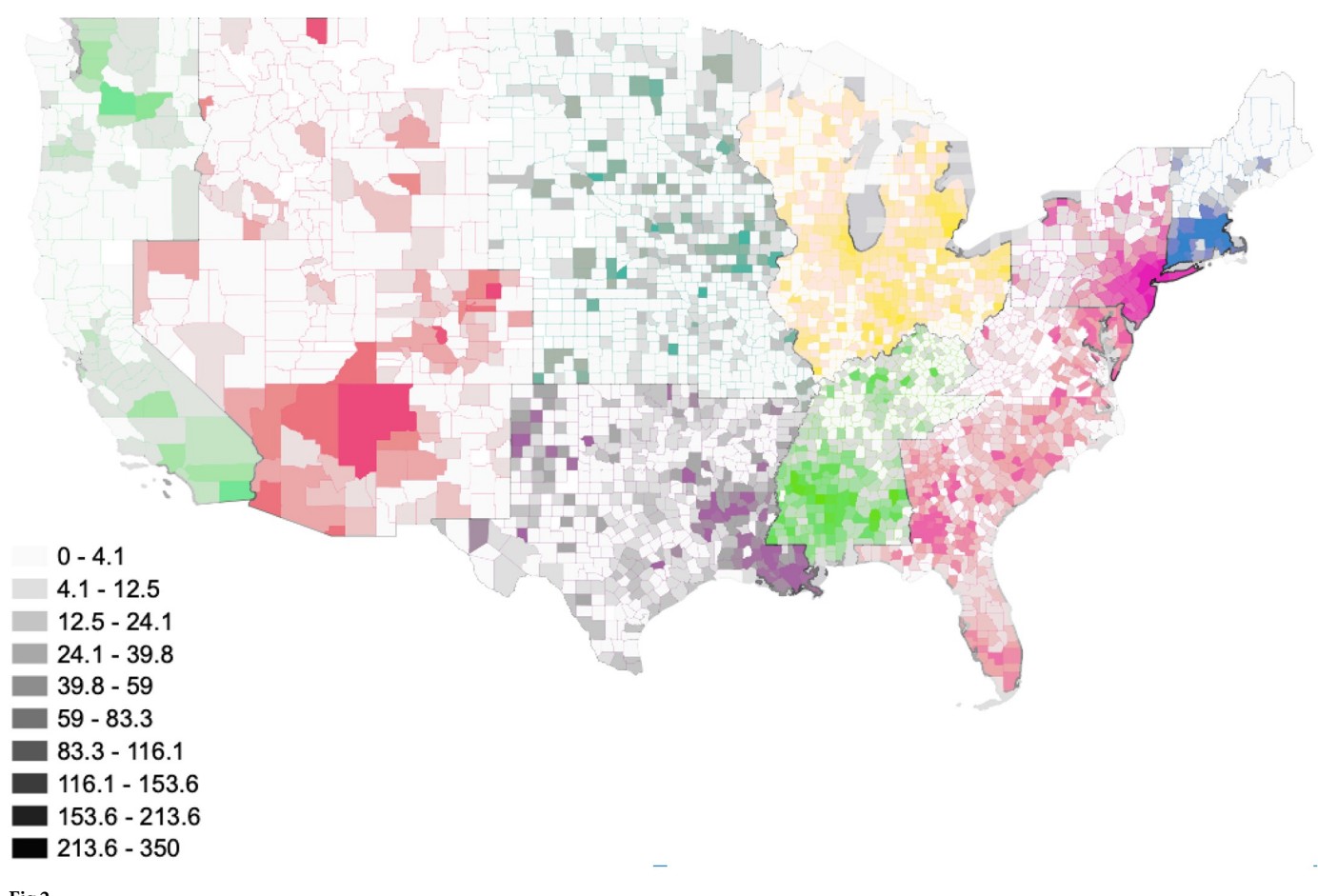

Legend:
- 0 - 4.1
- 4.1 - 12.5
- 12.5 - 24.1
- 24.1 - 39.8
- 39.8 - 59
- 59 - 83.3
- 83.3 - 116.1
- 116.1 - 153.6
- 153.6 - 213.6
- 213.6 - 350

**Fig 2.**

0.22 indirect, p <0.001). The percentage of residents over 65, number of residents per square mile, and county population were all also significantly associated with more deaths.

In urban counties (n = 115), the percentage of farmworkers and uninsured individuals were both significantly associated with more deaths, as was the percentage of residents over 65 and county population. Contrary to our initial hypotheses, in urban counties the percentage of uninsured individuals was associated with lower reported COVID-19 mortality. In these 115 counties, each percentage point decrease in the number of uninsured individuals was associated with 73.8 fewer reported COVID-19 deaths (p = 0.03). In rural areas, each increase in the percentage of uninsured individuals was associated with a direct effect of 0.69 fewer deaths within the county (p <0.001) and 0.19 fewer deaths in neighboring counties (p = 0.01). In non-urban counties (n = 2909), each percentage point increase in the number of farmworkers in a county was associated with 0.70 additional deaths (0.6 directly, 0.06 indirectly, p = 0.01). While neither the percentage of non-English speaking households nor the percentage of uninsured residents was associated with a higher number of reported deaths, each percentage point increase in poverty was associated with 0.49 additional deaths (0.4 directly, 0.04 indirectly, p = 0.04).

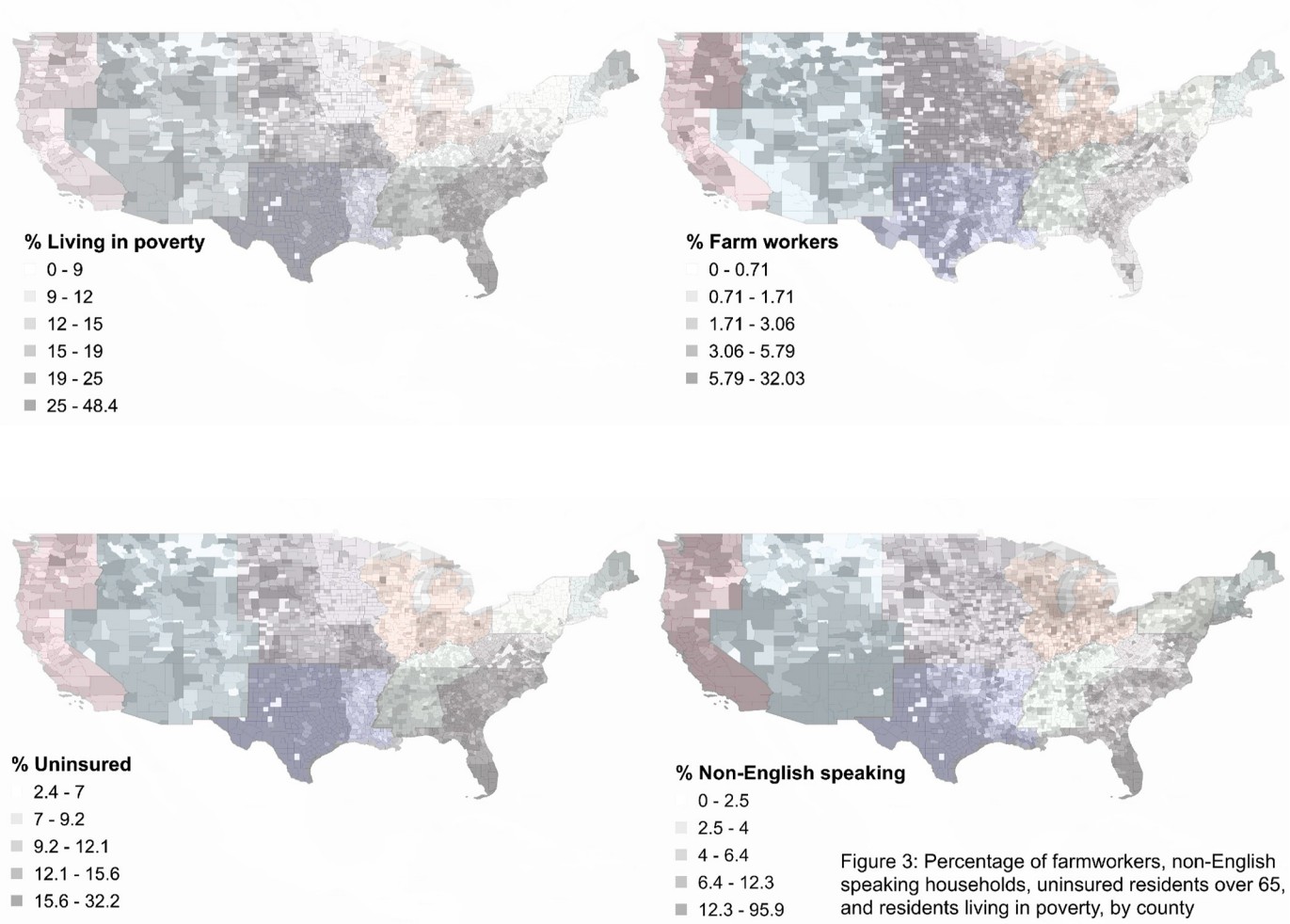

**Fig 3. Percentage of farmworkers, non-English speaking households, uninsured residents over 65, and residents living in poverty, by county.**

**Table 2. Full spatial regression models: Absolute number of deaths associated with each covariate for all counties and stratified by urban/rural.**

| | All counties (n = 3024) | | | | Non-urban counties (n = 2909) | | | | Urban counties (n = 115) | | | |
|---|---|---|---|---|---|---|---|---|---|---|---|---|
| | b-direct* | b-indirect | b-total | p-value | b-direct | b-indirect | b-total | p-value | b-direct | b-indirect | b-total | p-value |
| % Farm workers | 5.51 | 0.28 | 5.79 | 0.001 | 0.64 | 0.06 | 0.70 | 0.005 | 2280.53 | 258.98 | 2539.51 | 0.03 |
| % Non-English speakers | -2.77 | -0.14 | -2.92 | 0.83 | 0.09 | 0.01 | 0.10 | 0.24 | -13.18 | -1.50 | -14.67 | 0.24 |
| % Residents uninsured | 0.16 | 0.01 | 0.17 | 0.25 | -0.41 | -0.04 | -0.45 | 0.15 | -66.27 | -7.53 | -73.80 | 0.029 |
| % Residents in poverty | 4.20 | 0.22 | 4.41 | <0.001 | 0.44 | 0.04 | 0.49 | 0.04 | 4.47 | 0.51 | 4.98 | 0.86 |
| % Residents Over 65 | 4.36 | 0.22 | 4.58 | 0.002 | 0.67 | 0.07 | 0.73 | <0.001 | 0.30 | 0.03 | 0.34 | <0.001 |
| Residents per square mile | 0.24 | 0.01 | 0.25 | <0.001 | 0.08 | 0.01 | 0.08 | <0.001 | -15.76 | -1.79 | -17.55 | 0.75 |
| Population (thousands) | 0.62 | 0.03 | 0.65 | <0.001 | 0.23 | 0.02 | 0.25 | <0.001 | 1.22 | 0.14 | 1.36 | <0.001 |

*b-direct can be interpreted as the number of deaths associated with a given coviariate within a given county, b-indirect accounts for the spillover spatial effects on neighboring counties, and b-total can be interpreted as the total number of deaths associated with a covariate in a given county plus neighboring counties.

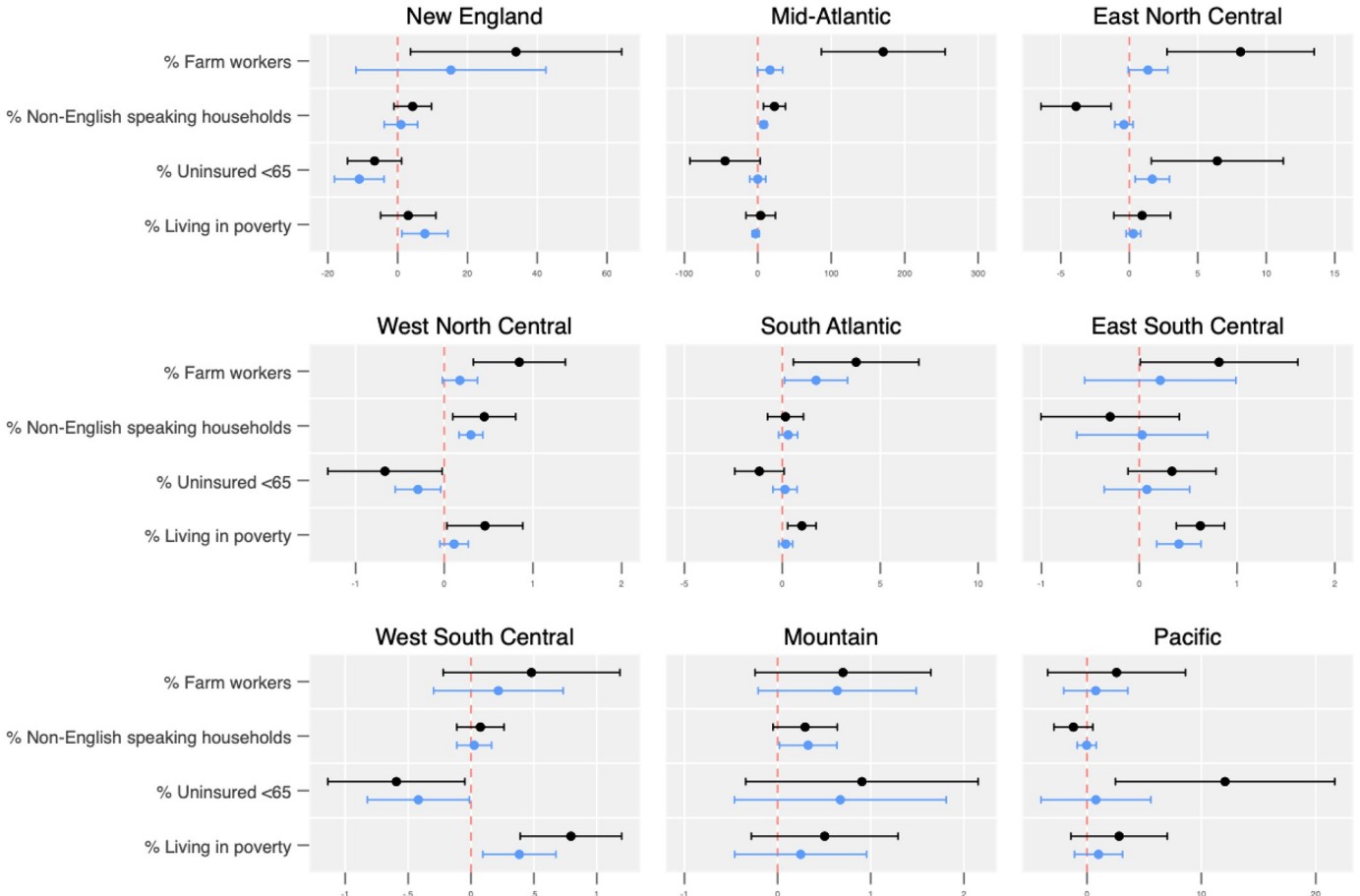

**Fig 4. Absolute number of deaths associated with covariates by US census region in all counties and non-urban counties.**

In sub-analyses by US Census region, distinct spatial patterns emerged (Fig 4). The percentage of farmworkers in a county was not significantly associated with more deaths in West South Central, Mountain, and Pacific states. A higher percentage of non-English speaking households was associated with a higher number of deaths in the Mid-Atlantic and West North Central regions. However, the same variable had a negative association with the number of COVID-19 deaths reported across all counties in the East North Central region. A higher percentage of uninsured individuals over the age of 65 is associated with a higher number of deaths in the East North Central, and Pacific region; however, the relationship is not significant in non-urban counties in the Pacific region, and the relationship is reversed in New England, the West North Central, and West South Central states, where a higher percentage of uninsured individuals is associated with fewer reported COVID-19 deaths.

When we assessed the rates of death per 100,000 individuals across the 9 census regions, we found similar risk patterns with one main exception: the percentage of farmworkers in a county was not significantly associated with number of deaths per 100,000 individuals in any region (Fig 5). Instead, the percentage of non-English speaking households in a county was significantly associated with higher rates of death across all counties in New England (total

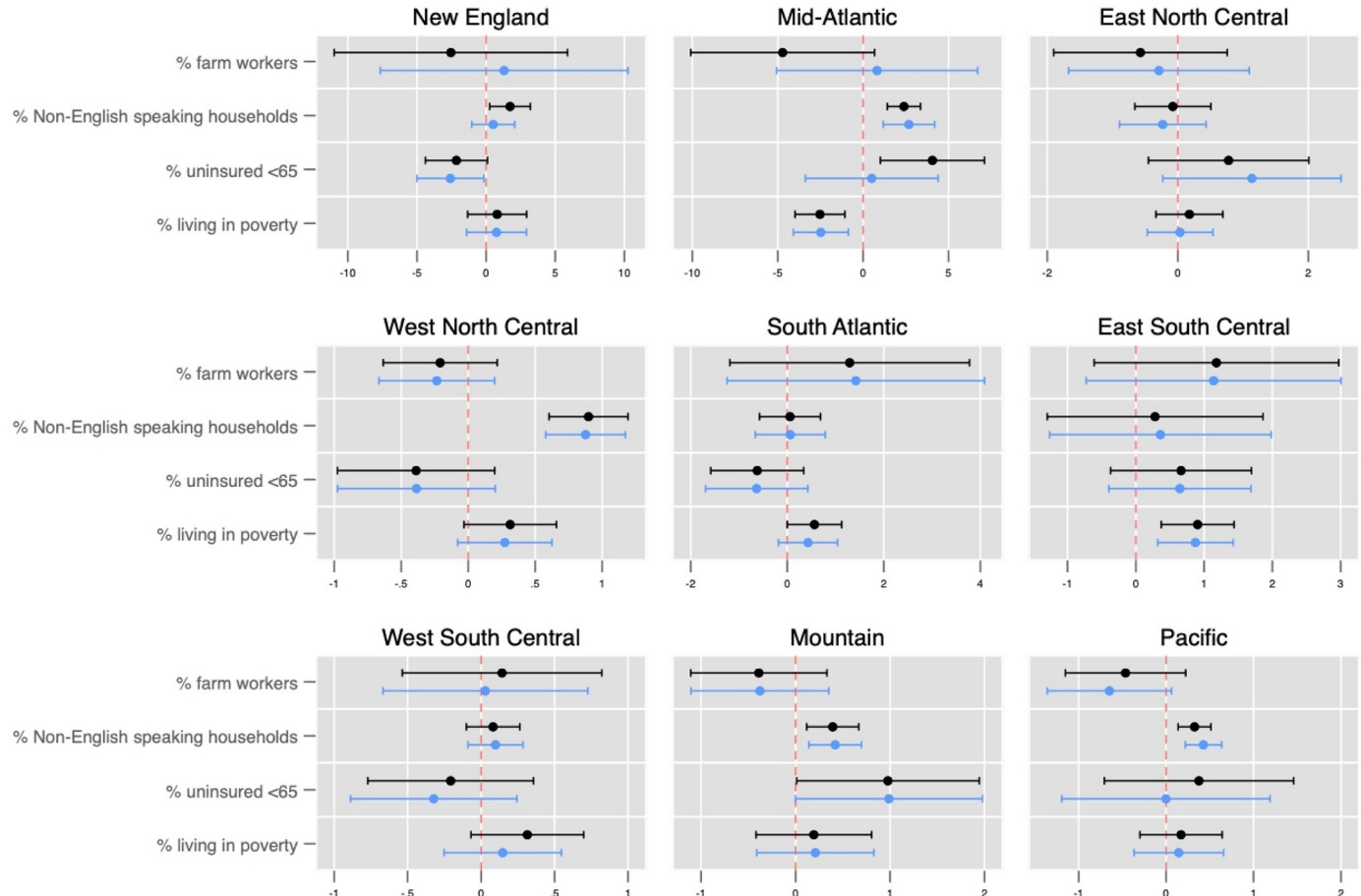

**Fig 5. Number of increased deaths per 100,000 residents associated with covariates in urban and non-urban counties by census region.**

b = 2.9, p = 0.02), the Mid-Atlantic (total b = 4.0, p < 0.001), and West North Central states (total b = 1.5, p < 0.001) and in non-urban counties in Mountain states (b = 1.20, p = 0.02). The percentage of uninsured individuals was associated with fewer reported COVID-19 deaths per 100,000 residents across all counties in New England (b = -3.6, p = 0.04) and in non-urban New England counties (b = -3.1, p = 0.02), but with higher rates across all counties in the Mid-Atlantic (b = 6.8, p = 0.01). Poverty was associated with 4.2 fewer reported deaths per 100,000 residents across all Mid-Atlantic counties (p <0.001) and 3.7 fewer reported COVID-19 deaths per 100,000 residents in non-urban Mid-Atlantic counties (p = 0.002), but with 4 more reported deaths per 100,000 residents in all counties in the East South Central region (p = 0.02) and 3.7 more reported deaths per 100,000 residents in non-urban counties in the same region (p = 0.03).

## Discussion

We used spatial autoregression models to assess the role of select social determinants of health as risk factors and drivers of the COVID-19 pandemic across the United States and by region.

Our findings highlight the fact that the US COVID-19 epidemic is better conceptualized as multiple simultaneous outbreaks across geographic regions rather than a single homogenous outbreak. The wide variety of responses and policy environments across states and regions has lead to corresponding variation in the role of socioeconomic factors such as insurance rates, poverty, and immigration status, as measured by the percentage of non-English speaking households.

The negative associations we found between poverty and mortality rates in the Mid-Atlantic and uninsured residents and mortality in New England are concerning. The CDC has noted higher than expected numbers of death across the United States in recent months, suggesting that COVID-19 mortality is potentially higher than what has thus far been captured by state and county level surveillance [17]. It is possible that these associations represent gaps in testing and linkage to care among immigrants and the uninsured, and/or a gap in ascertaining deaths due to COVID-19 among these same individuals.

The percentage of farmworkers in a county appears to be independently associated with a higher number of deaths, with the specific relationship varying by region. Across all 9 census regions, there was no association between the percentage of farmworkers and mortality when we used the number of deaths per 100,000 as our primary outcome, rather than the absolute number of deaths. Taken together, these two findings suggest that farmworkers may face unique risks of COVID-19 beyond issues of language, insurance, or economics, and that while their unique risk can be seen in higher absolute numbers of deaths, more farm workers in a county is not a major contributor to epidemic spread. Our finding that a higher percentage of non-English speaking households in a county is associated with a higher rate of deaths per 100,000 individuals does suggest that individuals who do not speak English may be at particularly high risk.

Farm labor is considered essential work, but there are reports of inadequate protections for this group of people, including inadequate personal protective equipment and inadequate social distancing guidelines, as well as a lack of enforcement [6, 18]. In addition to the risk to individual farmworkers, it is important to note that widespread outbreaks of COVID-19 among farmworkers also has the potential to impact food systems across the US. Although we cannot draw conclusions about individual risk profiles, our findings do suggest that farm work may create unique risk factors and that farmworkers may require additional protections, such as personal protective equipment and/or targeted outreach. Immigrants provide approximately 75% of all farm labor in the United States [8]. Among those engaged in crop work specifically, nearly three quarters are migrant workers, meaning that they travel from farm to farm during different growing seasons; and approximately half have undocumented citizenship status [8]. Undocumented status may impede an individual's willingness or ability to seek healthcare, or their ability to request additional protections from an employer if they worry doing so could result in their own deportation or that of a family member [19].

## Conclusion

COVID-19 mortality appears to be statistically significantly associated with social determinants of health at the county level, and these relationships may be more pronounced in non-urban counties. Individuals who do not speak English, individuals engaged in farm work, and individuals living in poverty may be at heightened risk for COVID-19 mortality in non-urban counties.

## Supporting information

**S1 File.**
(DOCX)

## Author Contributions

**Conceptualization:** Rebecca K. Fielding-Miller, Maria E. Sundaram.

**Data curation:** Rebecca K. Fielding-Miller.

**Formal analysis:** Rebecca K. Fielding-Miller.

**Funding acquisition:** Rebecca K. Fielding-Miller.

**Methodology:** Rebecca K. Fielding-Miller, Maria E. Sundaram, Kimberly Brouwer.

**Supervision:** Kimberly Brouwer.

**Validation:** Maria E. Sundaram.

**Visualization:** Rebecca K. Fielding-Miller.

**Writing – original draft:** Rebecca K. Fielding-Miller, Maria E. Sundaram.

**Writing – review & editing:** Rebecca K. Fielding-Miller, Maria E. Sundaram, Kimberly Brouwer.

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
