## [Decision Letter · Decision Letter 0]

30 Jun 2020

PONE-D-20-18322

Social determinants of COVID-19 mortality at the county level

PLOS ONE

Dear Dr. Fielding-Miller,

Thank you for submitting your manuscript to PLOS ONE. After careful consideration, we feel that it has merit but does not fully meet PLOS ONE’s publication criteria as it currently stands. Therefore, we invite you to submit a revised version of the manuscript that addresses the points raised during the review process.

We look forward to receiving your revised manuscript.

Kind regards,

Nickolas D. Zaller

Academic Editor

PLOS ONE

Journal Requirements:

2. To meet our reproducibility criteria, please ensure that the Methods section contains more information on data extraction; and that the data used is presented in supplementary tables.

3. We note you have included a table to which you do not refer in the text of your manuscript. Please ensure that you refer to Table 2 in your text; if accepted, production will need this reference to link the reader to the Table.

Additional Editor Comments (if provided):

As per reviewer #1, please consider conducting a sensitivity analysis using county population size to determine rates of mortality.

Reviewers' comments:

Reviewer's Responses to Questions

**Comments to the Author**

1. Is the manuscript technically sound, and do the data support the conclusions?

Reviewer #1: Yes

Reviewer #2: Yes

2. Has the statistical analysis been performed appropriately and rigorously? 

Reviewer #1: No

Reviewer #2: Yes

3. Have the authors made all data underlying the findings in their manuscript fully available?

Reviewer #1: Yes

Reviewer #2: Yes

4. Is the manuscript presented in an intelligible fashion and written in standard English?

Reviewer #1: Yes

Reviewer #2: Yes

5. Review Comments to the Author

Reviewer #1: This study provides an assessment of the associations between county-level factors and COVID-19 mortality, with an emphasis on the impact of higher concentrations of non-English speaking residents and residents who are farm workers. The study evaluates associations overall and by urban/non-urban status. The study provides valuable information for identifying which populations may be at high risk; however, the study would potentially benefit by addressing the comments below, with particular emphasis on considering evaluating a rate (e.g., per 1,000 residents) of COVID-19 mortality rather than a raw count of deaths.

Major

1. There is inconsistency in the lists of independent variables and associated findings, which causes confusion. For example, the abstract describes a significant finding with respect to “higher density” and urban status; however, this variable is not listed in the previous sentence as variables that were evaluated. It may be helpful to additionally clarify that other non-psychosocial variables were evaluated.

2. The word “more” is not necessarily accurate in the abstract (e.g., more farmworkers), as “more” relates to an absolute number rather than a relative number. The phrasing “a higher percentage of” may be more appropriate.

3. While I do not disagree with the content, I am not sure that the final sentence of the abstract is the best concluding sentence given other findings from the study.

4. Why were some counties (and/or county equivalents) excluded from the study? There are 3,142 counties and county equivalents in the 50 states, excluding DC. Based on the number of counties in the study, there are more counties excluded than if the difference were only based on excluding county equivalents. If this data were simply not available, that is all that needs to be stated (e.g., the data from XXX counties were unavailable).

5. I am curious as to why the analysis was not calculated based on rates of death (e.g., per 100,000 residents), rather than raw numbers. Given the range of number of individuals who live in counties, a value of 1 additional death may be relatively very different across counties, based on population size. While this is somewhat mitigated by separating based on urban and non-urban, the population sizes within these stratifications still vary substantially. For example, the population in LA county is greater than 10,000,000 versus the population in several urban counties in Virginia of only 5,000 residents. Note that some VA counties may have been excluded because of their county equivalency status, but even if so, the lowest non-VA county population is 69,000 residents (in Colorado). I think at least a sensitivity analysis using county population size to determine rates of mortality may be important additional information.

6. My understanding is that coefficients associated with the primary analyses indicate the association of a 1 unit (e.g., a 1 percentage point increase in percent uninsured) in a given variable “holding all else constant,” and the “constant” is at the means of each other variable. As such, additionally having averages displayed in Table 1 may be helpful for some readers.

7. It is my understanding that if a given model has an input variable that is on a scale from 0 to 100, the interpretation of the coefficient would be “a 1 percentage point increase in variable XX,” rather than a “percentage” increase (e.g., line 89).

8. I think it would be most helpful for the readers to have more descriptive column headings, associated with Table 2. For example, what exactly is “b direct.”

9. The first paragraph of the discussion is meaningful and should not be removed from the manuscript; however, the discussion would benefit from a first paragraph that first outlines the findings from the study.

Minor

10. “currently” (in first sentence of abstract and in Line 4 of the introduction) may be most appropriately replaced with the current date (e.g., “As of June 2020, the United States…”), to clarify for readers. For example, if an individual reads this in 2021 and there were to be another outbreak in a different country, this may be confusing for the reader. Relatedly, a more specific time frame should be associated with the first sentence of the introduction (e.g., “between January 2020 and June 2020”).

11. In the abstract when numbering the evaluated variables, the number “3” is used twice.

12. Add (“SIP”) for “shelter in place” in line 52.

13. Suggest “fitted the model” in line 71 be “fit the models”

14. IQR has a typo (either values switched or one number wrong) on line 81.

Reviewer #2: Incredibly well-written and thoughtful

Lines 17-22: a lot in that sentence. Could you break it down more, maybe 2 sentences. There is something about "language barrier" that seems judgy. What about "lack of multi-linguial public health communication" or something like that?

Line 39-43: I found this sentence describing the ACS survey confusing. Why did you use 2014 for one and 2017 for another?

Line 39: can you give 1-2 lines about how the NYT calculated this. Limitation: place of death may not be where they lived. Does the NYT report the deaths based on where they died or lived? I am guessing in rural counties, this is very important. There also has been some data about people going back and adjusting mortality with presumed COVID-19. Do you know if I tried to pull the same data today, it would be updated with different numbers?

Line 49: The calculation for "high risk days" is confusing to me. There is a high risk for contracting and then high risk for diagnosis. I agree, the risk of the virus contraction went down with the shelter order, but the risk of diagnosis did not go down until probably 2 weeks and then mortality 3-4 weeks after that. The mortality lags. I am not sure how important this variable is to your analysis, or if it will change it, but it is a definite weak point that should be addressed.

Lines 55-61: is there a reference that could support why you did this?

Line 65: would just write out Shelter in place order.

I found the table hard to read. why is deaths not capitalized? Could you put the non percent (residents per square mile) first or last to keep all the of % ones together? The title was also hard to understand. Is there a way you could make the table more readable? I am guessing a lot of the rural towns were very small, and that the data is reported by where they lived rather than the hospital based on the numbers

Line 91-103 and 110-120 so powerful. Well done.

Discussion

You start your discussion off apologizing, being humble. Why not just say Our findings suggest that farm work..." You have already discussed your methods, and individual risk profiles were not the goal. The first para of discussion 2 long. Can you shorten so that the reader will read 3-4 sentences that are really powerful, then go on to the next para

Line 142: "point of concern" is sort of wishy washy. It is not shocking. It is not new. Th negative association.... supports other literature that people who are poor experience barriers in accessing care."

6. PLOS authors have the option to publish the peer review history of their article (what does this mean?). If published, this will include your full peer review and any attached files.

Reviewer #1: No

Reviewer #2: **Yes: **Alysse Wurcel MD MS

---

## [Author Response · Author response to Decision Letter 0]

25 Aug 2020

Reviewer #1: This study provides an assessment of the associations between county-level factors and COVID-19 mortality, with an emphasis on the impact of higher concentrations of non-English speaking residents and residents who are farm workers. The study evaluates associations overall and by urban/non-urban status. The study provides valuable information for identifying which populations may be at high risk; however, the study would potentially benefit by addressing the comments below, with particular emphasis on considering evaluating a rate (e.g., per 1,000 residents) of COVID-19 mortality rather than a raw count of deaths.

Major

1. There is inconsistency in the lists of independent variables and associated findings, which causes confusion. For example, the abstract describes a significant finding with respect to “higher density” and urban status; however, this variable is not listed in the previous sentence as variables that were evaluated. It may be helpful to additionally clarify that other non-psychosocial variables were evaluated.

We have added the following language to the abstract and believe our methods are now clearer:

We further adjusted these models for total population, population density, and number of days since the first reported case in a given county.

2. The word “more” is not necessarily accurate in the abstract (e.g., more farmworkers), as “more” relates to an absolute number rather than a relative number. The phrasing “a higher percentage of” may be more appropriate.

This is a good point, and we appreciate the reviewer’s observation. We have made the suggested revision.

3. While I do not disagree with the content, I am not sure that the final sentence of the abstract is the best concluding sentence given other findings from the study.

We have revised the abstract extensively.

---

4. Why were some counties (and/or county equivalents) excluded from the study? There are 3,142 counties and county equivalents in the 50 states, excluding DC. Based on the number of counties in the study, there are more counties excluded than if the difference were only based on excluding county equivalents. If this data were simply not available, that is all that needs to be stated (e.g., the data from XXX counties were unavailable).

We were not sufficiently clear in describing our data source. The data reflect ever county in the United States in which at least one COVID-19 case has been reported as of the date the analyses were begun. The “missing” counties were those that had not yet reported a case of COVID-19. We have clarified this in the manuscript text. We have also updated our analyses with more recent data (as of July 13), and as a result our analyses contain a higher number of counties (n=3024)

5. I am curious as to why the analysis was not calculated based on rates of death (e.g., per 100,000 residents), rather than raw numbers. Given the range of number of individuals who live in counties, a value of 1 additional death may be relatively very different across counties, based on population size. While this is somewhat mitigated by separating based on urban and non-urban, the population sizes within these stratifications still vary substantially. For example, the population in LA county is greater than 10,000,000 versus the population in several urban counties in Virginia of only 5,000 residents. Note that some VA counties may have been excluded because of their county equivalency status, but even if so, the lowest non-VA county population is 69,000 residents (in Colorado). I think at least a sensitivity analysis using county population size to determine rates of mortality may be important additional information.

We thank the reviewer for this comment, and agree with their point. We reflected on this recommendation at length. Because the COVID-19 epidemic in the United States is perhaps better understood as a series of sub-epidemics with a great deal of spatial heterogeneity, we conducted a series of 9 sub-analyses looking at the rate of deaths per 100,000 by census region. We believe comparing these sub-analyses with our original models (which we have elected to retain, while adding county population as an additional independent variable) significantly enhances the richness of our study, and we thank are grateful for the suggestion.

6. My understanding is that coefficients associated with the primary analyses indicate the association of a 1 unit (e.g., a 1 percentage point increase in percent uninsured) in a given variable “holding all else constant,” and the “constant” is at the means of each other variable. As such, additionally having averages displayed in Table 1 may be helpful for some readers.

Thank you, we have added the median of each variable to Table 1 to assist the reader’s interpretation of our findings.

7. It is my understanding that if a given model has an input variable that is on a scale from 0 to 100, the interpretation of the coefficient would be “a 1 percentage point increase in variable XX,” rather than a “percentage” increase (e.g., line 89).

The reviewer is correct. We have modified the language accordingly throughout.

8. I think it would be most helpful for the readers to have more descriptive column headings, associated with Table 2. For example, what exactly is “b direct.”

We appreciate this point. We have added additional language to the body of the text to explain the meaning of b-direct and b-indirect. We understand that this analytic approach is not currently widely used, and that additional interpretation is necessary. In the interest of readability, we have decided to leave the explanation in the manuscript text and the headings as is, however at the editors discretion we are happy to modify.

9. The first paragraph of the discussion is meaningful and should not be removed from the manuscript; however, the discussion would benefit from a first paragraph that first outlines the findings from the study.

After reflecting at length on this comment and others, we have rewritten the discussion entirely. We believe it is much stronger now and appreciate the reviewer’s critique.

Minor

10. “currently” (in first sentence of abstract and in Line 4 of the introduction) may be most appropriately replaced with the current date (e.g., “As of June 2020, the United States…”), to clarify for readers. For example, if an individual reads this in 2021 and there were to be another outbreak in a different country, this may be confusing for the reader. Relatedly, a more specific time frame should be associated with the first sentence of the introduction (e.g., “between January 2020 and June 2020”).

The reviewer makes an excellent point, and we appreciate their confidence in future reader’s interest in this paper! We have modified the language as suggested and updated to the most recent numbers.

11. In the abstract when numbering the evaluated variables, the number “3” is used twice.

We have extensively revised the abstract and addressed this typo.

12. Add (“SIP”) for “shelter in place” in line 52.

At the suggestion of reviewer 2 we have removed this variable from our models. (see below)

13. Suggest “fitted the model” in line 71 be “fit the models”

We have made the suggested change.

14. IQR has a typo (either values switched or one number wrong) on line 81.

The numbers have been updated with more recent data and the typo addressed.

 

Reviewer #2: Incredibly well-written and thoughtful

Lines 17-22: a lot in that sentence. Could you break it down more, maybe 2 sentences. There is something about "language barrier" that seems judgy. What about "lack of multi-linguial public health communication" or something like that?

This is a fair point, and we appreciate the reviewer’s point that the barrier is in the lack of accessible health communications rather than vulnerable communities preferred language of communication. We have changed the line as follows: 

“…and that these risks are independent of poverty, insurance, or linguistic accessibility of COVID-19 health campaigns.

Line 39-43: I found this sentence describing the ACS survey confusing. Why did you use 2014 for one and 2017 for another?

We have revised this sentence for clarity. The 2014 5-year estimate was the most recent county-level data available from the ACS for the percentage of households in which “no one age 14 and over speaks English Only or Speaks English “very well”. More recent county-level data were available for the population age 65 and over in the United States. We have included citations for all tables in an attempt to clarify. 

Line 39: can you give 1-2 lines about how the NYT calculated this. Limitation: place of death may not be where they lived. Does the NYT report the deaths based on where they died or lived? I am guessing in rural counties, this is very important. There also has been some data about people going back and adjusting mortality with presumed COVID-19. Do you know if I tried to pull the same data today, it would be updated with different numbers?

This is such an important point, and we appreciate the reviewer’s comment. We have updated this section with further details on the NYT’s methodology, and with a more obvious link to the data for interested readers. The “patchwork nature” of COVID-19 reporting across the United States makes it difficult to offer a definitive answer to the reviewer’s question. An analysis conducted by Headwater Economics suggests that this issue may also differ geographically, with more hospital beds per capita in the western United States vs. Eastern. To address this and other regional differences we have conducted a second set of analyses by US region. (see response to reviewer 1, item 5). (https://headwaterseconomics.org/equity/hospital-access-seniors/)

Line 49: The calculation for "high risk days" is confusing to me. There is a high risk for contracting and then high risk for diagnosis. I agree, the risk of the virus contraction went down with the shelter order, but the risk of diagnosis did not go down until probably 2 weeks and then mortality 3-4 weeks after that. The mortality lags. I am not sure how important this variable is to your analysis, or if it will change it, but it is a definite weak point that should be addressed.

We agree, particularly given the different definitions of sheltering in place across counties and states. This variable has been removed, with no dramatic changes in model outcomes.

Lines 55-61: is there a reference that could support why you did this?

We have inserted the appropriate citation.

Line 65: would just write out Shelter in place order.

This variable has been removed from our analyses.

I found the table hard to read. why is deaths not capitalized? Could you put the non percent (residents per square mile) first or last to keep all the of % ones together? The title was also hard to understand. Is there a way you could make the table more readable? I am guessing a lot of the rural towns were very small, and that the data is reported by where they lived rather than the hospital based on the numbers

We have revised the tables somewhat for clarity. We have opted to leave the residents per square mile and population at the bottom of the table, as we wanted to showcase outcomes and predictors in descending order of theoretical importance to our models. We have also added additional figures showcasing some of the 

Line 91-103 and 110-120 so powerful. Well done.

Thank you

Discussion

You start your discussion off apologizing, being humble. Why not just say Our findings suggest that farm work..." You have already discussed your methods, and individual risk profiles were not the goal. The first para of discussion 2 long. Can you shorten so that the reader will read 3-4 sentences that are really powerful, then go on to the next para

We have extensively revised the discussion and hope it now addresses botht his point and the one below.

Line 142: "point of concern" is sort of wishy washy. It is not shocking. It is not new. Th negative association.... supports other literature that people who are poor experience barriers in accessing care."

We have changed this language.

---

## [Decision Letter · Decision Letter 1]

4 Sep 2020

PONE-D-20-18322R1

Social determinants of COVID-19 mortality at the county level

PLOS ONE

Dear Dr. Fielding-Miller,

Thank you for submitting your manuscript to PLOS ONE. While you have thoughtfully addressed many of the previous reviewer comments, some concerns remain. Therefore, we feel that your manuscript does not fully meet PLOS ONE’s publication criteria as it currently stands. We invite you to submit a revised version of the manuscript that addresses the points raised during the review process.

One of the more important concerns relates to the modeling approach and that there is a lack of detail regarding how the authors addressed (or did not address) the fact that a quarter of counties had a mortality equal to 0.  There are multiple statistical approaches that could be used to address potential modal instability due to a relatively large number of 0s.  At the very least, this should be further discussed in the limitations section of the manuscript. 

We look forward to receiving your revised manuscript.

Kind regards,

Nickolas D. Zaller

Academic Editor

PLOS ONE

Reviewers' comments:

Reviewer's Responses to Questions

**Comments to the Author**

1. If the authors have adequately addressed your comments raised in a previous round of review and you feel that this manuscript is now acceptable for publication, you may indicate that here to bypass the “Comments to the Author” section, enter your conflict of interest statement in the “Confidential to Editor” section, and submit your "Accept" recommendation.

Reviewer #1: (No Response)

Reviewer #2: All comments have been addressed

2. Is the manuscript technically sound, and do the data support the conclusions?

Reviewer #1: Partly

Reviewer #2: Yes

3. Has the statistical analysis been performed appropriately and rigorously? 

Reviewer #1: No

Reviewer #2: Yes

4. Have the authors made all data underlying the findings in their manuscript fully available?

Reviewer #1: Yes

Reviewer #2: Yes

5. Is the manuscript presented in an intelligible fashion and written in standard English?

Reviewer #1: Yes

Reviewer #2: Yes

6. Review Comments to the Author

Reviewer #1: This paper evaluates the association of COVID-19 mortality with social determinants at the county level. The authors have taken great efforts to address the comments by the reviewers, and the paper remains important in that it highlights aspects related to the COVID-19 pandemic. A few questions regarding method choices should be addressed as outlined below.

Major comments

1. The authors have taken the reviewer’s suggestion to evaluate rate of mortality per 100,000 residents stratified by census region. I suggest that the findings from this analysis be provided as well. Relatedly, I’m unsure if the information in lines 189-192 means overall or in the regional analysis. This could be a reasonable place to note the overall analysis per 100,000 in an appendix.

2. The authors should clearly address limitations of their study and what implications the limitations may have. For example, 25% of the counties included in the analysis have 0 deaths. With a highly skewed outcome (as evident by the range versus the IQR and median) it is important to highlight what this may mean in terms of the analysis using spatial autoregressive models. Second, there should be a discussion about the range of the primary variable of focus (percent of farmers) among urban counties, which has an IQR of 0.0 to 0.1. This may be related to the finding of an increase of 2,500 deaths for each 1 percentage point increase in urban counties. It is likely that this may not be a stable model.

3. Figures 4 and 5 may be a bit challenging to interpret without more context. Potentially change the words “percentage” to “a one percentage point change in.”

4. I am a bit confused on the map legends in Figure 3. There are multiple categories that show 0.0-0.0, which likely are extra categories or simple mislabeling/typos. However, I am unsure how the percent of residents in poverty are all in the hundreds or over 1,000.

Minor comments

1. There remains two more uses of “more” rather than “percent of” in the abstract and in the last sentence of the introduction. I suggest modifying these for consistency and accuracy

2. Median death count on line 103 does not match what is in table 1. Text median = 2, and table median = 6.

3. The population (thousands) variable is flipped between non-urban and urban counties in table 1.

4. In table 2, please clarify that these outcomes are related to the death counts analysis (rather than per 100,000). Potentially include a footer to explain what b-direct/indirect/total mean for the reader.

5. The highest end of the largest level in the legend in Figure 2 (1615.6) does not match the maximum given in the text 350).

Reviewer #2: Thank you for taking time to make a powerful analysis even sharper.

One thing:

Something off with line 186: "Even when adjusting for the percentage of non-186 English speaking households, percentage of people living in poverty, percentage of people without insurance, and the percentage of farmworkers in a county appears to be independently associated with a higher number of deaths, and that the relationship varies by region."

7. PLOS authors have the option to publish the peer review history of their article (what does this mean?). If published, this will include your full peer review and any attached files.

Reviewer #1: No

Reviewer #2: **Yes: **Alysse G. Wurcel MD MS

---

## [Author Response · Author response to Decision Letter 1]

14 Sep 2020

PONE-D-20-18322R1

Social determinants of COVID-19 mortality at the county level

PLOS ONE

Dear Dr. Fielding-Miller,

Thank you for submitting your manuscript to PLOS ONE. While you have thoughtfully addressed many of the previous reviewer comments, some concerns remain. Therefore, we feel that your manuscript does not fully meet PLOS ONE’s publication criteria as it currently stands. We invite you to submit a revised version of the manuscript that addresses the points raised during the review process.

One of the more important concerns relates to the modeling approach and that there is a lack of detail regarding how the authors addressed (or did not address) the fact that a quarter of counties had a mortality equal to 0. There are multiple statistical approaches that could be used to address potential modal instability due to a relatively large number of 0s. At the very least, this should be further discussed in the limitations section of the manuscript.

We have addressed this potential shortcoming under Reviewer 1, Comment 2. 

Comments to the Author

Reviewer #1: This paper evaluates the association of COVID-19 mortality with social determinants at the county level. The authors have taken great efforts to address the comments by the reviewers, and the paper remains important in that it highlights aspects related to the COVID-19 pandemic. A few questions regarding method choices should be addressed as outlined below.

Major comments

1. The authors have taken the reviewer’s suggestion to evaluate rate of mortality per 100,000 residents stratified by census region. I suggest that the findings from this analysis be provided as well. 

These findings are presented in lines 154 – 170. We present these findings by region rather than nationally because the national level findings would mask the significant variation by region.

Relatedly, I’m unsure if the information in lines 189-192 means overall or in the regional analysis. This could be a reasonable place to note the overall analysis per 100,000 in an appendix.

We have added language to this section to clarify that we are discussing sub-analyses. The line now reads as follows:

Across all 9 census regions, there was no association between the percentage of farmworkers and mortality…

2. The authors should clearly address limitations of their study and what implications the limitations may have. For example, 25% of the counties included in the analysis have 0 deaths. With a highly skewed outcome (as evident by the range versus the IQR and median) it is important to highlight what this may mean in terms of the analysis using spatial autoregressive models. Second, there should be a discussion about the range of the primary variable of focus (percent of farmers) among urban counties, which has an IQR of 0.0 to 0.1. This may be related to the finding of an increase of 2,500 deaths for each 1 percentage point increase in urban counties. It is likely that this may not be a stable model.

We appreciate the reviewer’s thoughtfulness. We agree, these data are highly non-normal and careful attention to model building is necessary. We have added the following language to the text regarding our approach: 

 Our spatial autoregressive model used the generalized spatial two-stage least squares estimator, specifically Stata’s spregress command with the “g2sl” option. G2sl is a generalized method of moment (GMM) modeling approach. The GMM makes no assumptions about variable distribution and is an appropriate statistical approach to address data that are highly skewed or have unknown distributions (15).

3. Figures 4 and 5 may be a bit challenging to interpret without more context. Potentially change the words “percentage” to “a one percentage point change in.”

We have revised the figures and hope they are now more clear.

4. I am a bit confused on the map legends in Figure 3. There are multiple categories that show 0.0-0.0, which likely are extra categories or simple mislabeling/typos. However, I am unsure how the percent of residents in poverty are all in the hundreds or over 1,000.

Thank you, some of these were errors from the software, others were typos. We have revised figure 3 extensively for clarity.

Minor comments

1. There remains two more uses of “more” rather than “percent of” in the abstract and in the last sentence of the introduction. I suggest modifying these for consistency and accuracy

2. Median death count on line 103 does not match what is in table 1. Text median = 2, and table median = 6.

Thank you for catching this oversight. We have corrected the text.

3. The population (thousands) variable is flipped between non-urban and urban counties in table 1.

Thank you! We appreciate the reviewer’s attention to detail. This has been corrected

4. In table 2, please clarify that these outcomes are related to the death counts analysis (rather than per 100,000). Potentially include a footer to explain what b-direct/indirect/total mean for the reader.

We appreciate this suggestion and agree that it will enhance the paper’s clarity.

5. The highest end of the largest level in the legend in Figure 2 (1615.6) does not match the maximum given in the text 350).

We have modified all figures to address the reviewer’s comments.

Reviewer #2: Thank you for taking time to make a powerful analysis even sharper.

One thing:

Something off with line 186: "Even when adjusting for the percentage of non-186 English speaking households, percentage of people living in poverty, percentage of people without insurance, and the percentage of farmworkers in a county appears to be independently associated with a higher number of deaths, and that the relationship varies by region.".

We have modified this sentence and believe it now reads more clearly.

---

## [Decision Letter · Decision Letter 2]

22 Sep 2020

Social determinants of COVID-19 mortality at the county level

PONE-D-20-18322R2

Dear Dr. Fielding-Miller,

We’re pleased to inform you that your manuscript has been judged scientifically suitable for publication and will be formally accepted for publication once it meets all outstanding technical requirements.

Kind regards,

Nickolas D. Zaller

Academic Editor

PLOS ONE

Additional Editor Comments (optional):

Reviewers' comments:

Reviewer's Responses to Questions

**Comments to the Author**

1. If the authors have adequately addressed your comments raised in a previous round of review and you feel that this manuscript is now acceptable for publication, you may indicate that here to bypass the “Comments to the Author” section, enter your conflict of interest statement in the “Confidential to Editor” section, and submit your "Accept" recommendation.

Reviewer #1: All comments have been addressed

2. Is the manuscript technically sound, and do the data support the conclusions?

Reviewer #1: Yes

3. Has the statistical analysis been performed appropriately and rigorously? 

Reviewer #1: Yes

4. Have the authors made all data underlying the findings in their manuscript fully available?

Reviewer #1: Yes

5. Is the manuscript presented in an intelligible fashion and written in standard English?

Reviewer #1: Yes

6. Review Comments to the Author

Reviewer #1: The authors have addressed all of the comments, and the manuscript provides important findings regarding the pandemic.

7. PLOS authors have the option to publish the peer review history of their article (what does this mean?). If published, this will include your full peer review and any attached files.

Reviewer #1: No

---

## [Editor Report · Acceptance letter]

28 Sep 2020

PONE-D-20-18322R2 

Social determinants of COVID-19 mortality at the county level 

Dear Dr. Fielding-Miller:

I'm pleased to inform you that your manuscript has been deemed suitable for publication in PLOS ONE. Congratulations! Your manuscript is now with our production department. 

Kind regards, 

on behalf of

Dr. Nickolas D. Zaller 

Academic Editor

PLOS ONE